# X2T: Training an X-to-Text Typing Interface with Online Learning from User Feedback

**Jensen Gao, Siddharth Reddy, Glen Berseth**
University of California, Berkeley
sgr@berkeley.edu

**Nicholas Hardy, Nikhilesh Natraj**
University of California, San Francisco

**Karunesh Ganguly**
University of California, San Francisco
karunesh.ganguly@ucsf.edu

**Anca D. Dragan, Sergey Levine**
University of California, Berkeley

## Abstract

We aim to help users communicate their intent to machines using flexible, adaptive interfaces that translate arbitrary user input into desired actions. In this work, we focus on assistive typing applications in which a user cannot operate a keyboard, but can instead supply other inputs, such as webcam images that capture eye gaze or neural activity measured by a brain implant. Standard methods train a model on a fixed dataset of user inputs, then deploy a static interface that does not learn from its mistakes; in part, because extracting an error signal from user behavior can be challenging. We investigate a simple idea that would enable such interfaces to improve over time, with minimal additional effort from the user: online learning from user feedback on the accuracy of the interface's actions. In the typing domain, we leverage *backspaces* as feedback that the interface did not perform the desired action. We propose an algorithm called *x-to-text* (X2T) that trains a predictive model of this feedback signal, and uses this model to fine-tune any existing, default interface for translating user input into actions that select words or characters. We evaluate X2T through a small-scale online user study with 12 participants who type sentences by gazing at their desired words, a large-scale observational study on handwriting samples from 60 users, and a pilot study with one participant using an electrocorticography-based brain-computer interface. The results show that X2T learns to outperform a non-adaptive default interface, stimulates user co-adaptation to the interface, personalizes the interface to individual users, and can leverage offline data collected from the default interface to improve its initial performance and accelerate online learning.

## 1 Introduction

Recent advances in user interfaces have enabled people with sensorimotor impairments to more effectively communicate their intent to machines. For example, Ward et al. (2000) enable users to type characters using an eye gaze tracker instead of a keyboard, and Willett et al. (2020) enable a paralyzed human patient to type using a brain implant that records neural activity. The main challenge in building such interfaces is translating high-dimensional, continuous user input into desired actions. Standard methods typically calibrate the interface on predefined training tasks for which expert demonstrations are available, then deploy the trained interface. Unfortunately, this does not enable the interface to improve with use or adapt to distributional shift in the user inputs.

In this paper, we focus on the problem of assistive typing: helping a user select words or characters without access to a keyboard, using eye gaze inputs (Ward et al., 2000); handwriting inputs (see Figure 7 in the appendix), which can be easier to provide than direct keystrokes (Willett et al., 2020); or inputs from an electrocorticography-based brain implant (Leuthardt et al., 2004; Silversmith et al., 2020). To enable any existing, default interface to continually adapt to the user, we train a model using online learning from user feedback. The key insight is that the user provides feedback on the interface's actions via *backspaces*, which indicate that the interface did not perform the desired

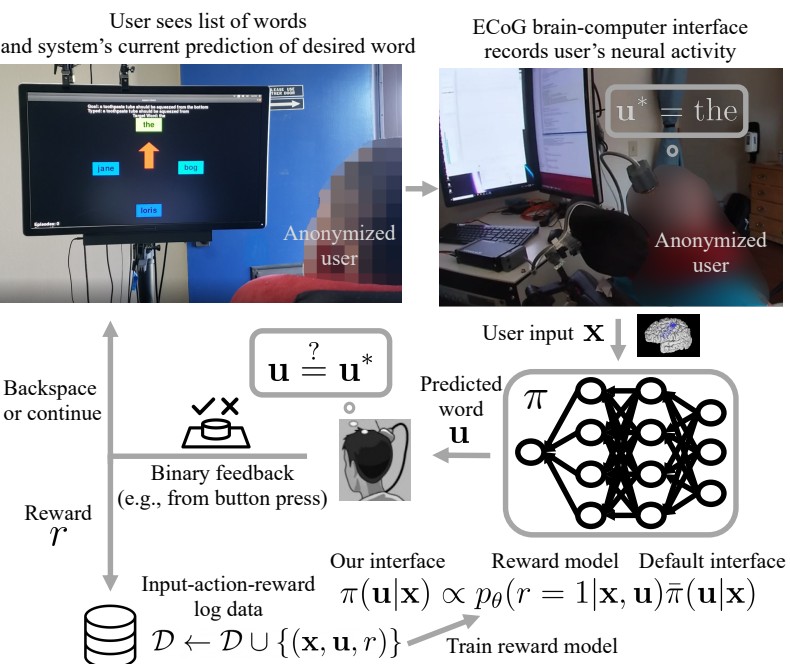

Figure 1: We formulate assistive typing as a human-in-the-loop decision-making problem, in which the interface observes user inputs (e.g., neural activity measured by a brain implant) and performs actions (e.g., word selections) on behalf of the user. We treat a backspace as feedback from the user that the interface performed the wrong action. By training a model online to predict backspaces, we continually improve the interface.

action in response to a given input. By learning from this naturally-occurring feedback signal instead of an explicit label, we do not require any additional effort from the user to improve the interface. Furthermore, because our method is applied on top of the user's default interface, our approach is complementary to other work that develops state-of-the-art, domain-specific methods for problems like gaze tracking and handwriting recognition. Figure 1 describes our algorithm: we initialize our model using offline data generated by the default interface, deploy our interface as an augmentation to the default interface, collect online feedback, and update our model.

We formulate assistive typing as an online decision-making problem, in which the interface receives observations of user inputs, performs actions that select words or characters, and receives a reward signal that is automatically constructed from the user's backspaces. To improve the default interface's actions, we fit a neural network reward model that predicts the reward signal given the user's input and the interface's action. Upon observing a user input, our interface uses the trained reward model to update the prior policy given by the default interface to a posterior policy conditioned on optimality, then samples an action from this posterior (see Figure 1). We call this method *x-to-text* (X2T), where *x* refers to the arbitrary type of user input; e.g., eye gaze or brain activity.

Our primary contribution is the X2T algorithm for continual learning of a communication interface from user feedback. We primarily evaluate X2T through an online user study with 12 participants who use a webcam-based gaze tracking system to select words from a display. To run ablation experiments that would be impractical in the online study, we also conduct an observational study with 60 users who use a tablet and stylus to draw pictures of individual characters. The results show that X2T quickly learns to map input images of eye gaze or character drawings to discrete word or character selections. By learning from online feedback, X2T improves upon a default interface that is only trained once using supervised learning and, as a result, suffers from distribution shift (e.g., caused by changes in the user's head position and lighting over time in the gaze experiment). X2T automatically overcomes calibration problems with the gaze tracker by adapting to the mis-calibrations over time, without the need for explicit re-calibration. Furthermore, X2T leverages offline data generated by the default interface to accelerate online learning, stimulates co-adaptation from the user in the online study, and personalizes the interface to the handwriting style of each user in the observational study.

## 2 LEARNING TO INFER INTENT FROM USER INPUT

In our problem setting, the user cannot directly perform actions; e.g., due to a sensorimotor impairment. Instead, the user relies on an assistive typing interface, where the user's intended action is inferred from available inputs such as webcam images of eye gaze or handwritten character drawings. As such, we formulate assistive typing as a contextual bandit problem (Langford & Zhang, 2008; Yue & Joachims, 2009; Li et al., 2010; Lan & Baraniuk, 2016; Gordon et al., 2019). At each timestep, the user provides the interface with a context $\mathbf{x} \in \mathcal{X}$, where $\mathcal{X}$ is the set of possible user inputs (e.g., webcam images). The interface then performs an action $\mathbf{u} \in \mathcal{U}$, where $\mathcal{U}$ is the set of possible actions (e.g., word selections). We assume the true reward function is unknown, since the user cannot directly specify their desired task (e.g., writing an email or filling out a form). Instead of eliciting a reward function or explicit reward signal from the user, we *automatically construct a reward signal from the user's backspaces*. The key idea is to treat backspaces as feedback on the accuracy of the interface's actions.

Our approach to this problem is outlined in Figure 1. We aim to minimize expected regret, which, in our setting, is characterized by the total number of backspaces throughout the lifetime of the interface. While a number of contextual bandit algorithms with lower regret bounds have been proposed in prior work (Lattimore & Szepesvári, 2020), we use a simple strategy that works well in our experiments: train a neural network reward model to predict the reward given the user's input and the interface's action, and select actions with probability proportional to their predicted optimality. Our approach is similar to prior work on deep contextual multi-armed bandits (Collier & Llorens, 2018) and NeuralUCB (Zhou et al., 2019), except that instead of using Thompson sampling or UCB to balance exploration and exploitation, we use a simple, stochastic policy.

### 2.1 MODELING USER BEHAVIOR AND FEEDBACK

Unlike in the standard multi-armed bandit framework, we do not get to observe an extrinsic reward signal that captures the underlying task that the user aims to perform. To address this issue, we infer rewards from naturally-occurring user behavior. In particular, in the assistive typing setting, we take advantage of the fact that we can observe when the user *backspaces*; i.e., when they delete the most recent word or character typed by the interface. To infer rewards from backspaces, we make two assumptions about user behavior: (1) the user can perform a backspace action independently of our interface (e.g., by pressing a button); (2) the user tends to backspace incorrect actions; and (3) the user does not tend to backspace correct actions. Hence, we assign a positive reward to actions that were not backspaced, and assign zero reward to backspaced actions. Formally, let $r \in \{0, 1\}$ denote this reward signal, where $r = 0$ indicates an incorrect action and $r = 1$ indicates a correct action.

### 2.2 TRAINING THE REWARD MODEL TO PREDICT FEEDBACK

In order to perform actions that minimize expected regret – i.e., the total number of backspaces over time – we need to learn a model that predicts whether or not the user will backspace a given action in a given context. To do so, we learn a reward model $p_\theta(r|\mathbf{x}, \mathbf{u})$, where $p_\theta$ is a neural network and $\theta$ are the weights. Since the reward $r \in \{0, 1\}$ can only take on one of two values, $p_\theta$ is a binary classifier. We train this binary classifier on a dataset $\mathcal{D}$ of input-action-reward triples $(\mathbf{x}, \mathbf{u}, r)$. In particular, we fit the model $p_\theta$ by optimizing the maximum-likelihood objective; i.e., the binary cross-entropy loss (see Equation 2 in the appendix).

Since X2T learns from human-in-the-loop feedback, the amount of training data is limited by how frequently the user operates the interface. To reduce the amount of online interaction data needed to train the reward model, we use offline pretraining. We assume that the user already has access to some default interface for typing. We also assume access to an offline dataset of input-action pairs generated by the user and this default interface. We assign zero rewards to the backspaced actions and positive rewards to the non-backspaced actions in this offline dataset, and initially train our reward model to predict these rewards given the user's inputs and the default interface's actions. Thus, when X2T is initially deployed, the reward model has already been trained on the offline data, and requires less online interaction data to reach peak accuracy.

---

**Algorithm 1** X-to-Text (X2T)

---

   **Require** $\bar{\pi}, \theta_{\text{init}}$          ▷ default interface, pretrained reward model parameters
   **while** true **do**
      $\mathbf{x} \sim p_{\text{user}}(\mathbf{x})$                            ▷ user gives input
      $\mathbf{u} \sim \pi(\mathbf{u}|\mathbf{x}) \propto p_\theta(r=1|\mathbf{x},\mathbf{u})\bar{\pi}(\mathbf{u}|\mathbf{x})$         ▷ interface performs action
      $r \leftarrow 0$ if user backspaces else 1             ▷ infer reward from user feedback
      $\mathcal{D} \leftarrow \mathcal{D} \cup \{(\mathbf{x},\mathbf{u},r)\}$           ▷ store online input-action-reward data
      $\theta \leftarrow \theta + \nabla_\theta \sum_{(\mathbf{x},\mathbf{u},r)\sim\mathcal{D}} \log\left(p_\theta(r|\mathbf{x},\mathbf{u})\right)$      ▷ update reward model w/SGD

---

## 2.3 Using the Reward Model to Select Actions

Even with offline pretraining, the initial reward model may not be accurate enough for practical use. To further improve the initial performance of our interface at the onset of online training, we combine our reward model $p_\theta(r|\mathbf{x},\mathbf{u})$ with the default interface $\bar{\pi}(\mathbf{u}|\mathbf{x})$. We assume that $\bar{\pi}$ is a stochastic policy and that we can evaluate it on specific inputs, but do not require access to its implementation. We set our policy $\pi(\mathbf{u}|\mathbf{x}) = p(\mathbf{u}|\mathbf{x}, r=1)$ to be the probability of an action conditional on optimality, following the control-as-inference framework (Levine, 2018). Applying Bayes' theorem, we get $p(\mathbf{u}|\mathbf{x}, r=1) \propto p(r=1|\mathbf{x},\mathbf{u})p(\mathbf{u}|\mathbf{x})$. The first term is given by our reward model $p_\theta$, and the second term is given by the default interface. Combining these, we get the policy

$$\pi(\mathbf{u}|\mathbf{x}) \propto p_\theta(r=1|\mathbf{x},\mathbf{u})\bar{\pi}(\mathbf{u}|\mathbf{x}). \tag{1}$$

This decomposition of the policy improves the initial performance of our interface at the onset of online training, and guides exploration for training the reward model. It also provides a framework for incorporating a language model into our interface, as described in Section 4.3.

Our x-to-text (X2T) method is summarized in Algorithm 1. In the beginning, we assume the user has already been operating the default interface $\bar{\pi}$ for some time. In doing so, they generate an 'offline' dataset that we use to train the initial reward model parameters $\theta_{\text{init}}$. When the user starts using X2T, our interface $\pi$ already improves upon the default interface $\bar{\pi}$ by combining the default interface with the initial reward model via Equation 1. As the user continues operating our interface, the resulting online data is used to maintain or improve the accuracy of the reward model. At each timestep, the user provides the interface with input $\mathbf{x} \sim p_{\text{user}}(\mathbf{x})$. Although standard contextual bandit methods assume that the inputs $\mathbf{x}$ are i.i.d., we find that X2T performs well even when the inputs are correlated due to user adaptation (see Section 4.2) or the constraints of natural language (see Section 4.4). The interface then uses the policy in Equation 1 to select an action $\mathbf{u}$. We then update the reward model $p_\theta$, by taking one step of stochastic gradient descent to optimize the maximum-likelihood objective in Equation 2. Appendix A.1 discusses the implementation details.

## 3 Related Work

Prior methods for training interfaces with supervised learning typically collect a dataset of input-action pairs, then fit a model that predicts actions given inputs. These methods tend to either assume access to ground-truth action labels from the user (Anumanchipalli et al., 2019; Wang et al., 2016; Karamcheti et al., 2020), or assume the user always intends to take the optimal action (Gilja et al., 2012; Dangi et al., 2013; 2014; Merel et al., 2015). Unfortunately, the user may not always be able to provide ground-truth action labels. Furthermore, in order for the system to compute optimal actions, the user must be instructed to perform specific calibration tasks for which the optimal policy is already known. These calibration tasks may not be representative of the tasks that the user intends to perform. This can lead to a distribution mismatch between the inputs that the model is trained on during the calibration phase, and the inputs that the model is evaluated on at test time when the user performs their desired tasks. Standard methods address this problem by periodically repeating the calibration process (Willett et al., 2020; Proroković et al., 2020), which can be time-consuming, disruptive, and requires assumptions about when and how frequently to re-calibrate. X2T overcomes these issues by continually learning from user feedback on tasks that naturally arise as the user types, rather than imposing separate training and test phases.

Extensive prior work on text entry systems (MacKenzie & Tanaka-Ishii, 2010) enables users to type using eye gaze (Ward et al., 2000), Braille (Oliveira et al., 2011), gestures (Jones et al., 2010), and

palm keyboards for wearable displays (Wang et al., 2015). X2T differs in that it enables the user to type using arbitrary inputs like eye gaze or handwriting, rather than a fixed type of input that restricts the flexibility of the system and must be anticipated in advance by the system designer.

X2T trains a typing interface through reinforcement learning (RL) with human-in-the-loop feedback instead of an explicit reward function. COACH (MacGlashan et al., 2017; Arumugam et al., 2019), TAMER (Knox & Stone, 2009; Warnell et al., 2017), and preference learning (Sadigh et al., 2017; Christiano et al., 2017) also train an RL agent without access to extrinsic rewards, but require explicit user feedback on the agent's behavior. X2T differs in that it learns from naturally-occurring feedback, which requires no additional effort from the user to train the agent. Other prior work trains RL agents from implicit signals, such as electroencephalography (Xu et al., 2020), peripheral pulse measurements (McDuff & Kapoor, 2019), facial expressions (Jaques et al., 2017; Cui et al., 2020), and clicks in web search (Radlinski & Joachims, 2006). X2T differs in that it trains an interface that always conditions on the user's input when selecting an action, rather than an autonomous agent that ignores user input after the training phase. Furthermore, X2T focuses on the assistive typing domain, where, to the best of our knowledge, backspaces have not yet been used to train an interface through RL. Related work on assistive robotic teleoperation interfaces proposes human-in-the-loop RL methods that minimally modify user actions (Pilarski et al., 2011; Broad et al., 2017; Reddy et al., 2018; Schaff & Walter, 2020; Du et al., 2020; Jeon et al., 2020). X2T differs in that it learns to operate on arbitrary types of user inputs, instead of assuming the user provides suboptimal actions.

## 4 EXPERIMENTAL EVALUATION

We seek to answer the following questions: **Q1** (Sec. 4.1): Does X2T improve with use and learn to outperform a non-adaptive interface? **Q2** (Sec. 4.2): Does the user adapt to the interface while the interface adapts to the user? **Q3** (Sec. 4.3): Does X2T personalize the interface to different input styles? **Q4** (Sec. 4.4): Do offline pretraining and an informative prior policy accelerate online learning? **Q5** (Sec. 4.4): Does online learning improve the interface beyond the initialization provided by offline pretraining? **Q6** (Sec. 4.5): Can X2T improve the accuracy of a brain-computer interface for typing? To answer **Q1-2**, we run a user study with 12 participants who use webcam images of their eyes to type via gaze (see Figure 8). To answer **Q3-5**, we conduct an observational study with prerecorded images of handwritten characters drawn by 60 users with a tablet and stylus (see Figure 7). To answer **Q6**, we conduct a pilot study with one participant using an electrocorticography-based brain-computer interface. In our experiments, we use default interfaces that are not necessarily the state of the art in gaze tracking or handwriting recognition, but are instead chosen to test the hypotheses in **Q1-6**. Appendix A.1 describes the experiment design in detail.

### 4.1 ADAPTING THE INTERFACE TO THE USER

In this experiment, we aim to test X2T's ability to improve over time, relative to a non-adaptive, default interface. To that end, we formulate a gaze-based word selection task in which we display a list of words to the user (see Figure 8), ask them to look at their desired word, record an image from their webcam that captures their eye gaze, and predict their desired word. To measure objective performance, we ask the user to type specific goal sentences. To simplify the experiment, we restrict the action space $\mathcal{U} = \{1, 2, 3, ..., 8\}$ to the eight buttons on the screen, and always assign the next word in the sentence to one of those eight buttons (see Figure 8).

We evaluate (1) a default interface that uses iTracker (Krafka et al., 2016) to estimate the user's 2D gaze position on the screen and select the nearest button, and (2) X2T. We calibrate the default interface once at the beginning of each experimental condition for each user, by asking the user to look at each of the eight buttons one by one, recording 20 eye image samples for each button in 2 cycles, and training a 2D gaze position estimator using the iTracker method. After calibration, the default interface stays fixed, and does not adapt to the user within the session; in part, because we do not know in advance which word the user intends to select, so we cannot automatically harvest the necessary paired data of gaze images and targets to re-calibrate the interface. Instead of periodically interrupting the user to gather new paired data and re-calibrate, X2T continually adapts to the user throughout the session, following Algorithm 1. Appendix A.1 describes the implementation of the default interface and X2T in further detail. We measure the performance of each method using the ground-truth accuracy of action selections. To access ground-truth actions for calculating accuracy,

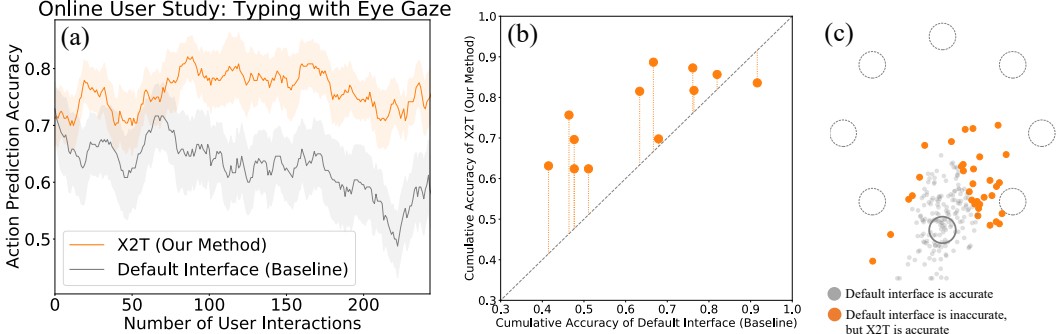

Figure 2: An online user study with 12 participants in the gaze tracking domain that addresses **Q1**: does X2T improve with use and learn to outperform a non-adaptive interface? **(a)** X2T predicts the user's intended action more accurately than the default interface, and the gap between the two methods grows over time. We smooth the curves using a moving average with a window size of 20 interactions, and measure standard error across the 12 users. **(b)** X2T improves the performance of 10 out of the 12 users, and the improvement from X2T is smaller when the default interface already performs well. Each orange circle represents one of the 12 users. The dashed gray line shows default-equivalent performance, and the dotted orange lines show the difference between X2T and default performance. Per-user accuracy is averaged across 250 interactions. **(c)** As shown in the screenshot in Figure 8, the user is shown a display of eight words arranged in a circle. Here, we plot the default interface's 2D gaze position estimates given user inputs intended to select the bottom-most button. The widely-scattered 2D estimates show that action prediction is particularly hard for this button, perhaps because the user's eyes tend to be more obscured when they are looking down. By training a reward model on user feedback, X2T helps the interface recover from incorrect gaze estimates.

we instruct the user to try to select a specified word from the list displayed on their screen. Note that this instruction is not an essential component of X2T, and is used purely to evaluate objective performance in this experiment.

The results in panel (a) of Figure 2 show that at the onset of online learning, both X2T and the default interface predict the user's intended action with the same accuracy, but quickly diverge. The default interface's performance degrades over time, potentially due to distribution shift in the user's inputs caused by changes in head position, background lighting, and other visual conditions. In contrast, X2T maintains the interface's strong initial performance throughout the experiment, by continually updating the reward model to reflect changes in the user's inputs. Panel (b) shows that X2T significantly improves the performance of 10 out of 12 participants, and that there are diminishing returns to X2T when the default interface already performs well. We ran a one-way repeated measures ANOVA on the action prediction accuracy dependent measure from the default and X2T conditions, with the presence of X2T as a factor, and found that $f(1,11) = 17.23, p < 0.01$. The subjective evaluations in Table 2 in the appendix corroborate these results: users reported feeling that X2T selected the words they wanted and improved over time more than the default interface. Panel (c) qualitatively illustrates how X2T helps the interface recover from incorrect 2D gaze position estimates: each green 'x' shows that even when the default interface estimates a gaze position far from the intended button, which would normally cause an incorrect action prediction, the reward model can adjust the overall action prediction back to the correct button via Equation 1. We also find that X2T performs well even when the user's feedback is slightly noisy: the user backspaces mistakes, and does not backspace correct actions, in 98.6% of their interactions.

### 4.2 USER ADAPTATION TO THE INTERFACE

In the previous experiment, we tested X2T's ability to adapt the interface to the user. Prior work on human-machine co-adaptation (Taylor et al., 2002; Carmena, 2013; Shenoy & Carmena, 2014; Nikolaidis et al., 2017) and the evolution of communication protocols between humans (Hawkins et al., 2020) suggests that an adaptive interface may not only learn to assist a static user, but even stimulate the user to adapt their input style to the interface. In this experiment, we investigate whether the user adapts to the gaze-based interface described in Section 4.1. To do so, we perform a counterfactual experiment: instead of training and evaluating X2T on inputs **x** generated by the user while they were typing with X2T, we train and

evaluate X2T on inputs **x** generated by the user while they were typing with the default interface. During the counterfactual experiment, instead of asking the user for new inputs, we simply replay the old inputs that were intended for the default interface, and automate backspaces.

This enables us to test if the user adapted their inputs to X2T, or if the user provided the same distribution of inputs to both the default interface and X2T.

The results in Figure 3 suggest that the user does indeed adapt their input style to the interface, and that this user adaptation improves performance. Comparing X2T's actual performance (orange curve) to X2T's counterfactual performance on replayed user inputs that the user originally intended for the default interface (teal curve), we see that X2T is able to perform much better with inputs intended for X2T compared to inputs intended for the default interface. From this result alone, one might infer that the user provided X2T with inputs that were more generically predictable than the inputs they provided to the default interface. However, by comparing the default interface's performance on replayed user inputs that the user originally intended for X2T (gray curve) to X2T's performance on the same inputs (orange curve), we see that X2T performs better than the default interface on the same inputs. This suggests that the user's inputs to X2T are not merely easier to predict, but in fact adapted specifically to X2T. In other words, X2T stimulates user co-adaptation to the interface.

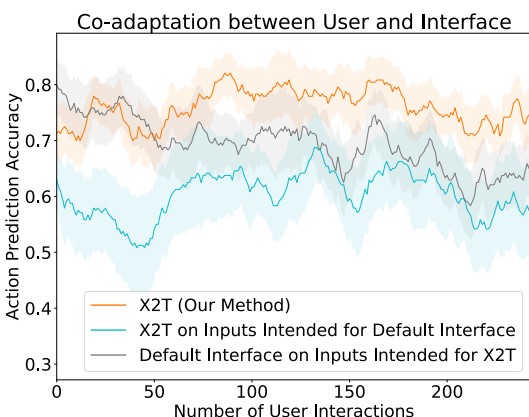

Figure 3: A counterfactual experiment with the online user study data that addresses **Q2**: does the user adapt to the interface while the interface adapts to the user? Training and evaluating X2T on user inputs originally intended for the default interface leads to worse performance than doing so on user inputs intended for X2T (orange vs. teal). Evaluating the default interface on inputs intended for X2T leads to worse performance than evaluating X2T on the same inputs (orange vs. gray). These results suggest that the user adapts their input style specifically to X2T, and that this user co-adaptation improves performance. We smooth the curves using a moving average with a window size of 20 interactions, and measure standard error across the 12 users.

## 4.3 PERSONALIZING THE INTERFACE FOR DIFFERENT USERS

In this experiment, we demonstrate X2T's ability to operate on a different type of user input: drawings of characters (see Figure 7 in the appendix); which, for some users, can be easier to provide than direct keystrokes (Willett et al., 2020). We also investigate to what extent X2T learns a personalized interface that is uniquely adapted to the individual user. To that end, we analyze handwriting samples from 60 users, collected through a tablet and stylus, from the UJI Pen Characters Database (Llorens et al., 2008). Each sample consists of a sequence of 2D positions that traces the trajectory of the user's pen tip as they draw a known character. We conduct an observational study with this data by sampling goal sentences, replaying a randomly-selected handwriting sample of the user's desired next character from the goal sentence, and treating each drawing as the user input **x**; akin to the replay experiment in Section 4.2. This observational study is equivalent to an online study, except that it does not permit user co-adaptation to the interface since it replays logged user inputs, and assumes that the feedback signal is not noisy when automating backspaces. To test X2T's robustness to noise and distributional shift in the user's input, we perturb the replayed pen tip positions by adding Brownian noise (see Figure 7 in the appendix).

We evaluate (1) a default interface trained to classify handwritten EMNIST characters (Cohen et al., 2017), and (2) X2T. We intentionally train the default interface on EMNIST images instead of UJI Pen Characters drawings, to model real-world applications with a distribution mismatch between the training data and test data, as discussed in Section 3. To address the challenge of selecting from 27 possible character actions, we use a language model (Dai et al., 2019) to predict the prior likelihood of an action $p_{\text{LM}}(\mathbf{u}_t|\mathbf{u}_{0:t-1})$ given the preceding characters $\mathbf{u}_{0:t-1}$ in the user's text field. We use this language model in both the default interface and X2T. In particular, we set $\bar{\pi}(\mathbf{u}|\mathbf{x}) \propto p_\phi(\mathbf{u}|\mathbf{x})p_{\text{LM}}(\mathbf{u}|\mathbf{u}_{0:t-1})$, where $p_\phi$ is the EMNIST image classifier.

As in Section 4.1, the default interface stays fixed throughout the experiment and does not adapt to the user, since re-training the default interface would require interrupting the user to collect paired data.

|          | Evaluation | | | |
|----------|------------|--------|--------|--------|
| Training | User 1 | User 2 | User 3 | User 4 |
| User 1 | **.995** | .033 | .025 | .017 |
| User 2 | .018 | **.463** | .009 | .061 |
| User 3 | .000 | .002 | **.975** | .039 |
| User 4 | .018 | .002 | .025 | **.993** |

Table 1: An observational study with 60 users in the handwriting recognition domain that addresses **Q3**: does X2T personalize the interface to different input styles? We measure action prediction accuracy across 1000 interactions, and randomly sample users 1-4 from the pool of 60 users. The interface trained on user $i$ is substantially more accurate when evaluated on inputs from user $i$ than on inputs from user $j$, suggesting that the learned interface is personalized to each individual user.

The results in Figure 4 show that X2T significantly outperforms the default interface (orange vs. gray). We ran a one-way repeated measures ANOVA on the action prediction accuracy dependent measure from the default and X2T conditions, with the presence of X2T as a factor, and found that $f(1, 59) = 37.46, p < 0.0001$. Furthermore, Table 1 shows that X2T learns an interface that is particularly suited to the user whose data it was trained on: when an interface trained on user $i$'s data is evaluated on data from user $j \neq i$ instead of user $i$, performance degrades substantially.

## 4.4 ABLATION EXPERIMENTS

In this experiment, we aim to test the importance of three components of X2T: offline pretraining, an informative prior policy, and online learning. As discussed in Sections 2.2 and 2.3, to improve the initial performance of X2T and accelerate online learning, we pretrain on offline data collected using the default interface, and incorporate prior knowledge from the default interface into the prior policy $\bar{\pi}$ in Equation 1. Using the handwriting recognition task from Section 4.3, we conduct ablation experiments in which we drop out each of the three components, one by one, and measure any resulting changes in performance. In the first condition, we test the effect of not performing offline pretraining, by initializing X2T with random weights $\theta_{\text{init}}$. In the second, we test the effect of not incorporating prior knowledge into our policy, by setting the prior policy $\bar{\pi}$ to be a uniform distribution over actions. In the third, we test the effect of not learning online and instead relying solely on offline pretraining, by freezing the reward model parameters after offline pretraining and not storing online data.

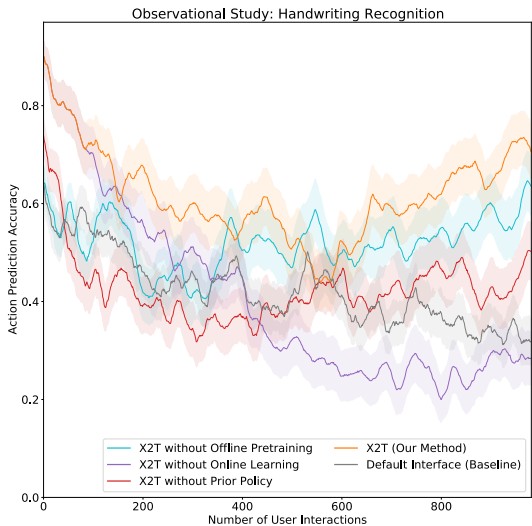

Figure 4: An observational study with 60 users in the handwriting recognition domain that addresses **Q4-5**: do offline pretraining and an informative prior policy accelerate online learning, and does online learning improve the interface beyond the initialization provided by offline pretraining? In this ablation experiment, we remove each of the three components – offline pretraining, an informative prior policy, and online learning – one by one, and find that each component is critical for maintaining high action prediction accuracy at different stages of the experiment.

The results in Figure 4 show that offline pretraining is helpful at the onset of online learning (orange vs. teal), but does not have a substantial effect on performance after some online data has been collected. This is unsurprising, since leveraging offline data is only necessary when insufficient online data has been collected. Using the default interface as a prior policy in Equation 1 is critical for X2T's performance throughout the experiment (orange vs. red). This is also unsurprising, given that the default interface contains useful prior knowledge about how to interpret user inputs. Online learning is not particularly helpful at the onset of the experiment (orange vs. purple), but has a substantial effect on performance over time. This result shows that learning from on-policy data is critical for X2T. In other words, X2T learns best when it learns from its own mistakes, rather than the mistakes of the default interface.

## 4.5 PILOT STUDY WITH A BRAIN-COMPUTER INTERFACE

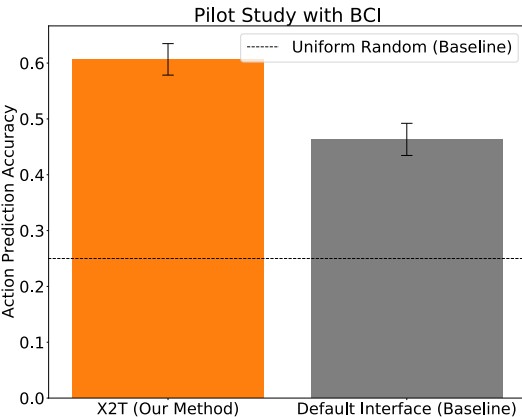

In this experiment, we demonstrate X2T's ability to improve the typing accuracy of a brain-computer interface (BCI) that uses a 128-channel chronic electrocorticography (ECoG) implant to measure neural activity (Leuthardt et al., 2004; Silversmith et al., 2020). Each user input has 128 features, which are obtained by pre-processing the raw ECoG signals (Appendix A.1.3 discusses the details). The results in Figure 5 show that, after offline pretraining on a dataset of 2913 input-action-reward tuples, X2T achieves 60.7% action prediction accuracy during a new session of 300 steps. We train a default interface on paired data of inputs and ground-truth actions collected before the offline data. To evaluate the default interface, we conduct a counterfactual experiment (similar to those in previous sections) on the 300 steps of data from the X2T evaluation session. The default interface only achieves 46.3% ac-

Figure 5: A pilot study with one participant that addresses **Q6**: can X2T improve the accuracy of a brain-computer interface (BCI) for typing? We find that X2T adapts to recent input-action-reward data and outperforms a default interface that is trained via supervised learning on older, paired input-action data.

curacy. Interestingly, the default interface never predicts the top-most button action, leading to a 0% recall rate for that action. This is most likely due to changes in the feature processing pipeline or other kinds of distributional shift in the user inputs over time. In contrast, X2T achieves 70.2% recall on the top-most button action, suggesting that it can overcome these challenges.

## 5 Discussion

In our online user study on gaze-based word selection, we show that X2T learns to outperform a non-adaptive interface in under 30 minutes (including offline data collection), and that the user simultaneously adapts their input style to the interface. The observational study on handwritten character recognition shows that X2T can also personalize the interface to individual users. Additionally, our pilot study with a brain-computer interface user shows that X2T improves the recall rate of one particular action from 0% to 70.2%. These experiments broadly illustrate how online learning from user feedback can be used to train a human-machine interface.

One limitation of this work is that we assume backspaces can be generated independently of X2T. This assumption restricts X2T to settings where the user can communicate a binary signal; e.g., through a button press or a sip-and-puff interface. One direction for future work is to relax this assumption by training a binary classifier to generate feedback signals from, e.g., facial expressions or brain activity. X2T is also limited in that the benefit from using X2T to fine-tune the default interface may decrease as the default interface is improved, as suggested by the diminishing returns in Figure 2. One direction for future empirical work is to test X2T with state-of-the-art default interfaces. In spite of these limitations, methods like X2T that learn from their mistakes and stimulate user co-adaptation provide a general mechanism for improving user interfaces; not only for assistive typing, but also for other domains, such as brain-computer interfaces for prosthetic limb control and augmented reality devices for visually-impaired users.

## 6 Acknowledgments

Thanks to Sarah Seko, Reza Abiri, Yasmin Graham, and members of the Neural Engineering and Plasticity lab at UC San Francisco for helping to conduct the brain-computer interface experiments in Section 4.5. Thanks to members of the InterACT and RAIL labs at UC Berkeley for feedback on this project. This work was supported in part by an NVIDIA Graduate Fellowship, AFOSR FA9550-17-1-0308, NSF NRI 1734633, NIH New Innovator Award [1 DP2 HD087955], and UCSF Weill Institute for Neurosciences.

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

# A  APPENDIX

## A.1  IMPLEMENTATION DETAILS

**Stochastic gradient descent.** We use Adam (Kingma & Ba, 2014) to optimize the binary cross-entropy loss described in Section 2.2,

$$\ell(\theta) = - \sum_{(\mathbf{x}, \mathbf{u}, r) \in \mathcal{D}} r \log \left( p_\theta(r = 1 | \mathbf{x}, \mathbf{u}) \right) + (1 - r) \log \left( 1 - p_\theta(r = 1 | \mathbf{x}, \mathbf{u}) \right). \tag{2}$$

We set gradient norm clipping to 10 in all experiments.

**Offline pretraining.** To pretrain X2T on offline data, we first train the reward model $p_\theta$ to convergence on the offline data, and store the learned network weights $\theta_{\text{init}}$. Once a minimum of four online input-action-reward triples have been collected, after every interaction, we update the reward model $p_\theta$ by taking one gradient step. We use the values $\theta_{\text{init}}$ to initialize the network weights (e.g., instead of a random initialization).

### A.1.1  ONLINE USER STUDY: TYPING WITH EYE GAZE

**Eye image features.** Instead of operating directly on 224x224 images of the user's eyes, we use the activations of the last fully-connected layer in the iTracker model (Krafka et al., 2016) as the input $\mathbf{x}$ to our reward model $p_\theta(r | \mathbf{x}, \mathbf{u})$. When training the reward model $p_\theta$, we freeze the iTracker network weights used to generate this 128-dimensional input $\mathbf{x}$.

**Reward model architecture and learning.** We set the learning rate for Adam to $10^{-3}$, and batch size to 128. We do not perform online updates to the reward model parameters $\theta$ while $|\mathcal{D}| < 4$. In our experiments, for every input $\mathbf{x}$, there is exactly one desired action $\mathbf{u}^*$ that will result in a positive reward $r = 1$, while all other actions $\mathbf{u} \neq \mathbf{u}^*$ result in a zero reward $r = 0$. In other words, $p(r = 1 | \mathbf{x}, \mathbf{u}) = p(\mathbf{u} = \mathbf{u}^* | \mathbf{x})$. Hence, we structure the reward model for X2T as $p_\theta(r = 1 | \mathbf{x}, \mathbf{u}) = f_\theta(\mathbf{u} | \mathbf{x})$, where $f_\theta$ is an action classifier. Furthermore, since we expect the reward model to learn to implicitly estimate the user's gaze position in order to predict actions, we directly incorporate this inductive bias into the model: we structure the action classifier as $f_\theta(\mathbf{u} | \mathbf{x}) \propto \exp \left( - \| g_\theta(\mathbf{x}) - \text{pos}(\mathbf{u}) \|_2 \right)$, where $g_\theta(\mathbf{x})$ outputs a 2D position of the user's estimated gaze, and $\text{pos}(\mathbf{u})$ is the known 2D position of the button for action $\mathbf{u}$. Note that even though $g_\theta(\mathbf{x})$ outputs a 2D position, the parameters $\theta$ are still trained on the reward prediction objective in Equation 2 (e.g., instead of a 2D gaze position prediction objective). We represent $g_\theta$ using a feedforward neural network with one hidden layer containing 64 units, a dropout layer with a dropout rate of 0.3 between the hidden layer and the output layer, and ReLU activations. At each timestep $t$, we record 10 eye images $\{\mathbf{x}_t^i\}_{i=1}^{10}$ at a sampling rate of 10 Hz, and average our predictions over these 10 inputs. Specifically, for X2T, we set $g_\theta(\mathbf{x}_t) = \frac{1}{10} \sum_{i=1}^{10} g_\theta(\mathbf{x}_t^i)$. For the default interface, we average the 2D gaze position estimates across the 10 samples before predicting the action whose button position is nearest to the average gaze position estimate. We initialize $\theta_{\text{init}}$ with the offline pretraining scheme described in Section 2.2, using 250 input-action-reward triples collected with the default interface.

**Experiment design.** We recruited 11 male and 1 female participants, with an average age of 21. Each participant was provided with the rules of the task and a short practice period of 20 interactions to familiarize themselves with the system. Each interaction – which consisted of providing an input, observing the interface's action, and deciding whether or not to backspace – took an average of 4 seconds. Each participant completed three phases of experiments: A, B, and C. In phase A, they operate the default interface for 250 steps, generating an offline dataset of input-action-reward triples that we use to initialize X2T. In phase B, they operate X2T for 250 steps. In phase C, they operate the default interface for 250 steps. To avoid the confounding effects of user learning or fatigue over time, we counterbalance the order of phase B and C: six randomly-selected participants completed phase B before C, and the other six participants completed phase C before B. Phase A is used solely to generate offline data to initialize X2T in phase B. One participant's room lighting changed substantially during phase B. Since their performance during phase A was substantially better than during phase B, we use their phase A data (instead of phase B data) to measure the default interface's performance on this one participant. We sample goal sentences from the MOCHA-TIMIT database (Wrench, 1999), following prior work on speech interfaces (Makin et al., 2020). We set the same goal sentences for each user in each condition.

**Deterministic policy.** In our experiments, we find that sampling actions $\mathbf{u} \sim \pi(\mathbf{u}|\mathbf{x})$ from the stochastic policy $\pi$ does not substantially improve exploration, and can in fact degrade performance by not always choosing the optimal action. This is most likely due to the use of the default interface $\bar{\pi}$ in Equation 1, which already provides an effective exploration mechanism. Hence, instead of randomly sampling actions, we deterministically select the highest-likelihood action: $\mathbf{u} \leftarrow \arg\max_{\mathbf{u}} \pi(\mathbf{u}|\mathbf{x})$.

### A.1.2 OBSERVATIONAL STUDY: TYPING BY DRAWING CHARACTERS

**Perturbing user inputs.** We perturb the character drawings in the UJI Pen Characters Database by decomposing each drawing into a sequence of pen tip velocity vectors, adding Brownian noise to each velocity, then integrating over the perturbed velocities to yield a complete, perturbed drawing (see Figure 7 in the appendix). We compute the Brownian noise by sampling an independent Gaussian noise vector with zero mean and variance of $2 \cdot 10^{-4}$ at each timestep, and summing over these noise vectors from time 0 to time $t$ to compute the Brownian noise vector for time $t$. The same Brownian noise vector is applied to all the velocity segments in a given drawing. Each user input $\mathbf{x}$ is a 28x28 image of the user's complete, perturbed drawing of a given character. There are 27 possible characters in the action space $\mathcal{U}$: 26 lower-case letters, and space (which we represent using the digit 7).

**Reward model architecture and learning.** We set the learning rate for Adam to $5 \cdot 10^{-4}$, batch size to 128, pretrain on the offline data for 20 epochs, and sample actions from the stochastic policy described in Equation 1 (instead of the deterministic policy described in Appendix A.1.1). We also limit the size of the replay buffer $\mathcal{D}$ in Algorithm 1 to the latest 500 input-action-reward triples. We do not perform online updates to the reward model parameters $\theta$ while $|\mathcal{D}| < 100$. We represent the reward model $p_\theta$ as a neural network with the following architecture: 28x28 input layer, 32x5x5 convolutional layer, 2x2 max pool layer, dropout layer with dropout rate of 0.5, 64x5x5 convolutional layer, 2x2 max pool layer, dropout layer with dropout rate of 0.3, and a fully-connected output layer. We structure the reward model as an action classifier, as in Section A.1.1. We initialize $\theta_{\text{init}}$ with the offline pretraining scheme described in Section 2.2, using 1000 input-action-reward triples collected with the default interface.

**Experiment design.** The UJI Pen Characters Database (Llorens et al., 2008) contains handwriting samples from 60 users. For each user, it includes two repetitions of lowercase letters, uppercase letters, and 10 digits, for a total of 1364 samples per user. In our observational study, we randomly sample target sentences, and replay user inputs that attempt to type the characters in those target sentences. In the replays, we automatically backspace incorrect actions; a realistic modeling choice, given that in the online user study in Section 4.1, the user did indeed backspace mistakes, and did not backspace correct actions, in 98.6% of their interactions. As in the online user study, we run each user's data through two conditions: default and X2T. Since this is an observational study, we do not need to counterbalance the order of the two conditions. For the personalization experiment in Table 1, we assign a randomly-selected, constant value to the random seed of the Brownian noise for each user in each condition. This ensures that comparisons between entry $(i, i)$ and entries $(i, j \neq i)$ in the 4x4 table are not confounded by differences in the input noise, and are only influenced by systematic differences in the users' individual handwriting styles. As in the online user study, we sample goal sentences from the MOCHA-TIMIT database (Wrench, 1999), and set the same goal sentences for each user in each condition.

**Language model.** We use the Transformer-XL language model (Dai et al., 2019) – specifically, the word-level version that is pretrained on the One Billion Word dataset (Chelba et al., 2013) – to compute the prior likelihood $p_{\text{LM}}(\mathbf{u}_t|\mathbf{u}_{0:t-1})$. To compute character-level likelihoods, we marginalize over the 60000 words in the language model's vocabulary that occur the most frequently in the One Billion Word training corpus, not including words with punctuation and converting upper-case to lower-case characters. We feed the previously-typed characters in the current sentence (i.e., not including the previous sentences) as context to the language model.

### A.1.3   Pilot Study with a Brain-Computer Interface

Due to constraints on the duration and number of sessions we were able to conduct with the participant, we made two simplifications to the interface: we simplified the display to use 4 buttons instead of 8 buttons, and we automated backspaces.

**Reward model architecture and learning.** We set the learning rate for Adam to $10^{-4}$, batch size to 128, and maximum buffer size $|\mathcal{D}|$ to 1000. Unlike the previous experiments, we did not use dropout. We used an L2 regularization constant of 0.001 for X2T, and 0.01 for the default interface. For both the default interface and reward model, we used a feedforward network architecture with 3 layers of 256 hidden units each. We deterministically selected actions with the maximum conditional likelihood under the policy, instead of sampling from the stochastic policy in Equation 1. When pretraining the reward model on offline data, we initialized the reward model network weights with the default interface network weights. We set the prior policy $\bar{\pi}$ in Equation 1 to be a uniform prior, instead of using the default interface. To accommodate noise in the user inputs, we require 4 consecutive bins of input to be mapped to the same action by the policy before executing that action; after 10 bins with no string of 4 consecutive, equal actions, the majority action is executed. In the counterfactual experiment with the default interface, if we reach the end of the bins for a given step without a string of 4 consecutive, equal actions, the majority action is executed. After an action is executed and reward collected, we add an input-action-reward tuple for the input at each bin to the buffer $\mathcal{D}$.

**Experiment design.** We conducted three experimental sessions: the first on 2/26/21, in which we collected 997 input-action-reward samples; the second on 3/5/21, which yielded 1916 samples; and the third on 3/13/21, which yielded 1984 samples. The default interface was trained on 3686 samples (left button intended: 959, down: 916, right: 897, up: 914) recorded prior to 2/26/21. There were 3 seconds of no input before each word selection. The interface was paused intermittently during each session to accommodate user fatigue.

**Feature processing.** The ECoG signals were binned at a frequency of 2 Hz. Each user input has 128 dimensions, which consist of high-band frequencies of raw ECoG signal recorded during each binning period. The remaining implementation details are described in Silversmith et al. (2020).

|  | $p$-value | Default Interface | X2T |
|---|---|---|---|
| The system selected the words I wanted | $< .05$ | 4.50 | **5.42** |
| The system improved over time | $< .05$ | 3.17 | **4.75** |
| I improved at using the system over time | $> .05$ | 4.08 | 4.42 |
| The system did not select the words I wanted | $< .05$ | 4.08 | **2.83** |
| The system got worse over time | $> .05$ | 3.25 | 2.42 |
| I got worse at using the system over time | $> .05$ | 3.25 | 2.67 |
| I backspaced when there was a mistake | $> .05$ | 5.42 | 5.83 |
| I ignored mistakes (did not backspace them) | $> .05$ | 2.33 | 2.17 |

Table 2: Subjective evaluations from the 12 participants in the online user study. Means reported below for responses on a 7-point Likert scale, where 1 = Strongly Disagree, 4 = Neither Disagree nor Agree, and 7 = Strongly Agree. $p$-values from a one-way repeated measures ANOVA with the presence of X2T as a factor influencing responses.

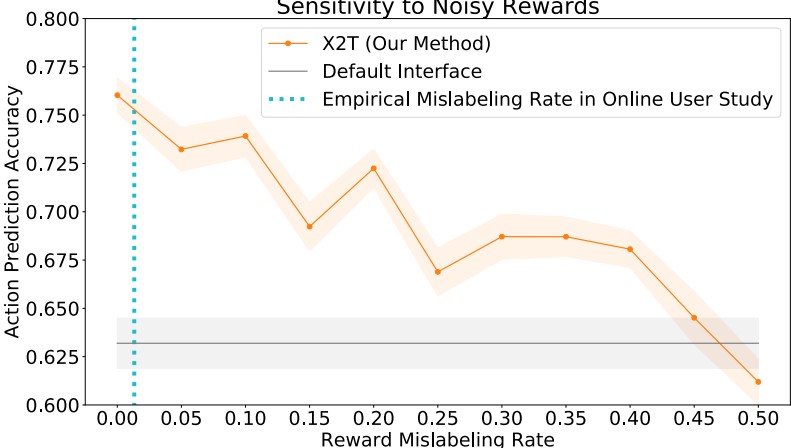

Figure 6: To measure X2T's sensitivity to noise in the reward signal, we conduct a counterfactual experiment with the eye gaze user study data, similar to Section 4.2. In this experiment, we flip the reward $r$ to $1 - r$ with probability $p$, which we call the reward mislabeling rate. We then train our reward model on these noisy rewards. X2T outperforms the default interface for a wide range of mislabeling rates, and only performs worse than the default interface when the rewards are completely random (i.e., the mislabeling rate is 50%). In practice, we find that users' backspaces tend to follow the assumptions in Section 2.1, which leads to a relatively low empirical mislabeling rate of 1.4%.

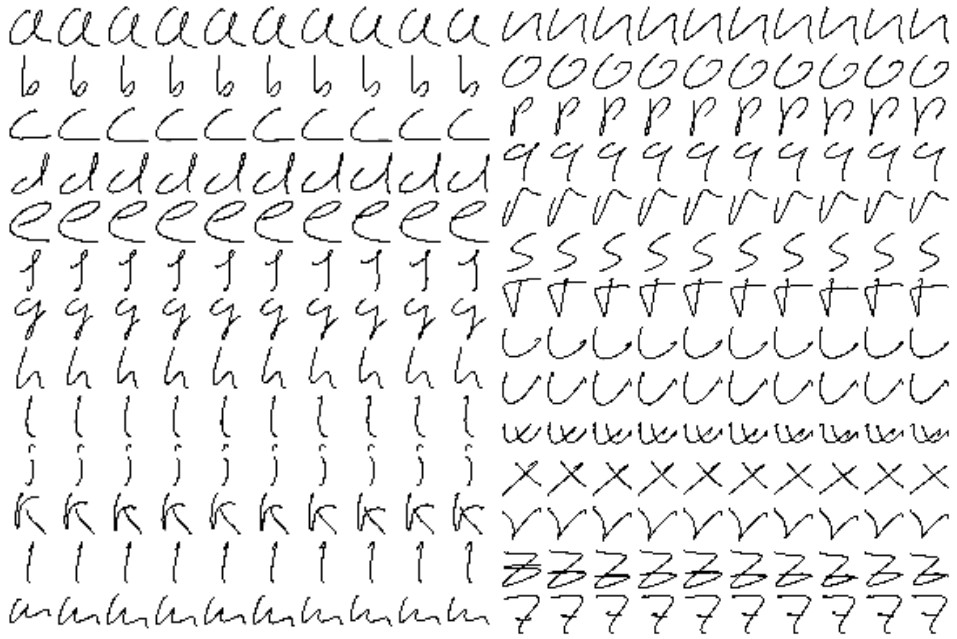

Figure 7: Handwriting samples from the UJI Pen Characters Database that have been perturbed by adding Brownian noise to pen tip velocities. The left-most column shows the true input, and each successive column shows the perturbed input after 100-timestep intervals. We used the character '7' in place of the space character.

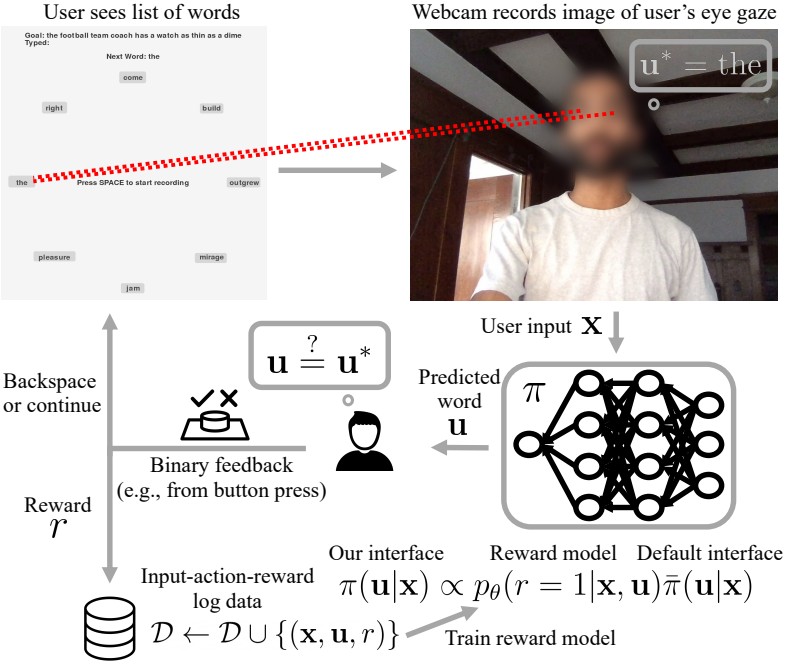

Figure 8: An illustration of the eye gaze experiments in Section 4.1

