# OpenReview forum: "X2T: Training an X-to-Text Typing Interface with Online Learning from User Feedback"
_ICLR.cc/2021/Conference — ICLR 2021 Poster_

### Official Review · AnonReviewer1 · 2020-10-27
**An attractive and practical algorithm for improving user interfaces**

**Rating:** 7
**Confidence:** 4

**Review:**

SUMMARY:
The authors propose a simple algorithm for using online learning from implicit human feedback to improve systems that operate in the contextual bandit setting. The main idea is to capture the presence/absence of corrective actions and use this information to infer a reward signal that the system can use to make decisions later. The proposed method is instantiated for text-entry tasks as a system called XT2, and the authors perform extensive empirical evaluation that seems to show that the system is very successful.


STRENGTHS:
	(S1) The authors present a simple-yet-seemingly-powerful algorithm for building adaptive interfaces from implicit user feedback.
	(S2) The proposed algorithm can easily leverage existing interfaces and therefore provides an interesting path to improving all kinds of existing systems.
	(S2) The authors instantiate the algorithm in a compelling use-case scenario -- non-traditional text entry -- and demonstrate seemingly compelling results empirically.


WEAKNESSES:
	(W1) It's unclear as to whether or not the evaluation in the major plots ("[Number of User Interactions] vs. [Action Prediction Accuracy]") is meaningful. In particular, it's not clear what data was used to compute the reported prediction accuracy. It would seem that this is computed over the prediction *for that time step*, but then more detail is needed on the datasets in order to determine whether or not the task somehow got easier for XT2 over time instead of it actually learning.
	(W2) Some of the experimental results are confusing. For example, what is the explanation for the position of the red circles in Figure 2c? There seems to be very little intuitive reasoning behind where red vs. green appears -- has XT2 really learned such a complicated and sensitive decision surface? If yes, why should we "trust" this decision surface? In analyzing 2c closely, it seems like a nearest-neighbor classifier would have done just as well.
	(W3) While I greatly appreciate the authors making clear their assumptions about backspaces in Section 2.1, the authors did not justify why their first assumption ("the user can perform a backspace action independently of our interface") is valid. It would seem to me that, in situations where XT2 is necessary, that such an assumption may not actually be valid.
	(W4) In Section 2.2, the authors state that they "assign a positive reward to actions that were not backspaced," but this choice is also not justified. It seems to me that there could be cases in which the user simply accepts an incorrect action from the system without providing the feedback.


RECOMMENDATION STATEMENT:
While I have listed a number of weaknesses above, overall I really like the paper. The algorithm is simple and can build upon existing interfaces as opposed to simply replacing them. Moreover, the algorithm is built to leverage data that is easy to collect, and therefore it seems feasible that similar systems could be deployed for a number of applications. That said, the current paper is lacking in the description of the experimental results and justifications for some of the assumptions.


QUESTIONS FOR AUTHORS:
	(Q1) How was accuracy computed for the "[Number of User Interactions] vs. [Action Prediction Accuracy]" figures?  How did the authors ensure that the task didn't somehow get easier regardless of system learning?
	(Q1) What justification do the authors have for their first assumption about backspaces (i.e., that they can be performed independently -- and, presumably, reliably -- of the XT2 interface).
	(Q2) What justification do the authors have for their implicit assumption that no backspace means positive reward?
	(Q3) While Table 1 is convincing that XT2 adapts to individual users, it also brings up the question as to how a model trained on _all_ user data would do. Do the authors have the ability to comment on this?


MINOR COMMENTS:
	(MC1) I find it odd that the system is named "x-to-text" but abbreviated "XT2." Should it not be "X2T" since one would replace the "to" with "2" ("x 2 text" -> "X2T")? I'm not taking any issue with the current name if this is how the authors meant it, rather just wondering if this is a typo.

---

> ### Author Response · Authors · 2020-11-14
> **Response to R1**
>
> Thank you for the thoughtful and constructive feedback.
>
> (Q1a) R1 is correct that we compute accuracy for each individual timestep, so that we can measure how accuracy changes over the course of the experiment. It is unlikely that the task became easier for XT2 over time, since the default interface performed worse over time, which suggests that the task may have actually been getting more difficult due to distributional shift in the user’s inputs.
>
> (Q1b) R1 and R3 both raise this important concern about the assumption that backspaces can be generated independently of XT2. This assumption restricts XT2 to settings where the user can communicate a binary signal through a single-button press or a sip-and-puff interface, or in settings where a separate binary classifier has already been trained to generate feedback signals from facial expressions or brain activity. We will add this discussion to Section 5.
>
> (Q2) R1 is correct that, in practice, users may choose not to backspace even when the previous action was incorrect, leading to noisy rewards. To address this concern raised by R1 and the other reviewers, we have added a new experiment (Figure 5 in the appendix) that shows XT2 outperforms the default interface even when the reward signal is noisy, although XT2’s accuracy does decrease as the reward misabeling rate increases.
>
> (Q3) In this setting, training a single model on all users’ data would probably perform worse than training a separate model on each individual user’s data, since each user’s handwriting style is different. Assuming the users’ input styles are similar, we could potentially use meta-learning to leverage data from previous users to accelerate learning from a new user.
>
> (W2) We apologize for the miscommunication. Our reward model does not take the default interface’s 2D gaze position estimates as input, but rather the original webcam image as input. Hence, XT2 can outperform the default interface, even when the default interface’s 2D gaze position estimates are inaccurate (the green ‘x’s in the scatterplot). The red circles indicate that while XT2 tends to outperform the default interface, there are still a few instances where XT2 predicts the incorrect action when the default interface would have predicted the correct action. We will clarify this in the caption.
>
> (MC1) XT2 was intended abbreviate x-to-text in the same way that TD3 abbreviates Twin Delayed Deep Deterministic policy gradient [1], but we actually like R1’s suggestion to use X2T instead, and will consider changing the acronym in the final paper.
>
> [1] https://arxiv.org/abs/1802.09477

---

> > ### Comment · AnonReviewer1 · 2020-11-24
> > **Follow Up**
> >
> > Thanks to the authors for their response; I'm satisfied with the answers and the updates to the manuscript.
> >
> > I've read the threads with the other reviewers, and I'll keep my score as-is at a 7. With regard to R2's concerns in general, I'd recommend the authors moderate the claims in the paper in order to make clear that their solution is particular to the studied domains only and that other approaches in the literature are probably more appropriate for more complex domains.

---

> > > ### Author Response · Authors · 2020-11-24
> > > **Response to follow up**
> > >
> > > Thank you for your comments and suggestions. We have moderated the claims made in the 2nd paragraph of Section 2 to clarify that more complex domains may require more sophisticated exploration strategies, such as the gap-dependent exploration method mentioned by R2. We have also updated the future work in Section 5 to include testing X2T with complementary methods for training personalized language models, such as the Gmail Smart Compose algorithm mentioned by R2.

---

### Official Review · AnonReviewer3 · 2020-10-28
**Official Blind Review #3**

**Rating:** 8
**Confidence:** 4

**Review:**

Summary:
This work presents a method for online learning of an assistive typing user interface (XT2) with implicit user feedback. User inputs for such an assistive typing interface are assumed to be in the form of eye gaze or handwritten characters. However, the implicit human feedback is assumed to be backspaces typed on a keyboard. Backspaces are used to delete words predicted by the assistive typing interface based on the user’s input. The online learning of such an interface to improve its assistive performance and adapt to the user over time is framed as a contextual bandit problem. A reward prediction network is trained to predict the use of backspaces (implicit feedback) by the user. This reward prediction network combined with the default interface policy using Baye’s theorem is used to update the policy of the typing interface. The experimental results with two user studies reveal that the presented method performs better than a non-adaptive default interface, stimulates user co-adaptation to the interface, and offline learning accelerates online learning.

########################


Pros:
- The paper is well-written and easy to follow.
- Strong experimental results to support the idea of the work.
- Offline pretraining of the reward model reduces the burden on online user interactions needed for the interface to be improved.
- The presented technique is applicable to any form of user inputs and the authors test their approach with two different forms of inputs (eye gaze and handwritten characters).
- Use of backspaces to train an interface with RL is a novel idea.

########################

Cons:
- The implicit feedback is assumed to be perfect and available via a keyboard. How realistic is this assumption? What happens if the feedback is noisy? Is the availability of an independent backspace action being true to the nature of the interface being truly assistive?
- It was unclear to me while reading the paper if the implicit feedback (backspaces) are provided via a keyboard or are also predicted with the user’s input (such as with gaze tracking). I assumed its the former with subtle hints in the paper and Fig 1. It would be helpful to make this explicit for the reader in Section 2.1 (as part of assumption 1).
- This approach does not model the temporal effects of learning. Practically, modeling the problem as an MDP would be more realistic versus contextual bandits.

########################

Reason for score:
This simple approach is presented with clarity and supported with well-reported experiments (including several ablative analyses). Some minor issues in the writing could be improved but overall the idea is well-presented and well-evaluated.

########################

Questions during rebuttal:
- Sec 4.1: Is the user study conducted with the interface type variable being varied within-subjects (i.e. each user uses both default and X2T)? From the experimental results, it seems this is the case but it is not explicitly stated. If this is so, is the order in which users attempt to use the two interfaces (default and X2T) counterbalanced?
- Please address other questions raised as part of the Cons section and other feedback.

########################

Some typos and other feedback:
- Section 2,  paragraph 1, sentence 2: “… relies on an assistive typing interface to infer the user’s intended action from available inputs …” -> “… relies on an assistive typing interface, where the user’s intended action is inferred from available inputs …”
- Algorithm 1: Consider defining what p_user(x) is. Are you making any assumptions on such a model of human user inputs (such as random sampling as suggested in Sec 4.3)?
- Consider citing a recent work on learning from imperfect implicit user feedback such as facial expressions): Cui, Y., Zhang, Q., Allievi, A., Stone, P., Niekum, S., & Knox, W. B. (2020). The EMPATHIC Framework for Task Learning from Implicit Human Feedback. Conference on Robot Learning (CoRL), 2020.
- Even though compared to prior work, this work does not assume access to ground truth action labels from the user provided to the interface, it does assume access to ground truth backspace actions. It would be beneficial to emphasize this in paragraph 1 of Section 3.
- It only becomes clear to me by Section 3 what is meant by “handwriting as an input” and how it can be an assistive input modality. A reference to Appendix Figure 5 early on the introduction, along with highlighting this can be an easier mode of user input versus typing on a keyboard, would be helpful.
- Section 4, paragraph 1: References to the subsection numbers can be made when stating the evaluation questions. For example: Q1 (Sec 4.1): Does X2T improve with use and learn to outperform a non-adaptive interface? The questions are well-framed and very clear though!
- Section 42: One of the insights presented for the presented method is that the XT2 interface can learn to automatically overcome calibration issues with the gaze tracker, thus the interface adapts to the mis-calibrations over time without the need for recalibration, even though external conditions would require a recalibration for better eye gaze prediction. This should be highlighted in the introduction as well.
- Section 4.3: Isn’t p_LM(u) conditioned on the preceding characters of the text seen so far? Would p_LM(u|t) be a better representation?
- Section 4.4, last paragraph: Consider reversing the order of the first two results presented (they are in opposite order to the questions posed in the previous paragraph).

---

> ### Author Response · Authors · 2020-11-14
> **Response to R3**
>
> Thank you for the thoughtful and constructive feedback.
>
> Addressing cons:
>  - We have added a new experiment (Figure 5 in the appendix) that shows XT2 outperforms the default interface even when the reward signal is noisy, although XT2’s accuracy does decrease as the reward misabeling rate increases. R3 raises an important concern about the assumption that backspaces can be generated independently of XT2. This assumption restricts XT2 to settings where the user can communicate a binary signal through a button press or a sip-and-puff interface, or in settings where a separate binary classifier has already been trained to generate feedback signals from facial expressions or brain activity. We will add this discussion to Section 5.
>  - We have updated assumption (1) in Section 2.1 as per R3’s suggestion.
>  - We agree with R3 that the contextual bandit assumption is limiting, and that formulating assistance as an MDP is more general. In this project, we chose to trade off generality for sample efficiency, since learning a single-step reward model requires less data than learning a sequential policy. We will clarify this decision in Section 5.
>
> Addressing questions:
>  - We used a within-subjects allocation and counterbalanced the order of the two conditions. Appendix A.1.1 includes additional details about the experiment design. We will move these details into the main paper.

---

> > ### Comment · AnonReviewer3 · 2020-11-20
> > **Follow up by R3**
> >
> > Thanks for your response and updated experiment to show how noise in the reward signal impact's XT2's performance. I have some follow-up questions and comments:
> > - Can the authors clarify the domain on which Fig 5 in the appendix was evaluated? I am assuming this is the gaze-based interface, based on the reference to Section 4.2 in the caption of Figure 5. However, it would be helpful for the user if this was made explicit.
> > - Fig 5: Is the noise in the reward signal added prior to training the reward prediction model or in the output of the predicted rewards?
> > - Fig 5: How many user interactions were used for Fig 5? Was this kept fixed for this experiment? If so, how was that value chosen?
> > - Similar to concerns raised by other reviewers, the assumption that providing implicit feedback via a keyboard is not exactly how an assistive interface would work. This feedback should not be called "implicit" and the discussion/introduction section must clearly address how implicit feedback can increase the complexity of the problem being attempted in this work. The authors should not run the risk of overselling the paper with more than what it can offer and demonstrate.
> > - That said, I do like the thoroughness of the experiments, evaluations with 2 human subject studies, and clarity of the paper.
> > - I agree with R1 that X2T would be a better acronym for the title.
> > - Please also address any typing errors and suggested recommendations by all reviewers for the main text of the paper.

---

> > > ### Author Response · Authors · 2020-11-20
> > > **Updates in response to R3's suggestions**
> > >
> > > Thank you again for the constructive feedback.
> > >
> > >  - Yes, we used the eye gaze user study data in Figure 5. We have updated the caption to clarify this.
> > >  - The noise is added to the rewards before training the reward model. We have added this information to the caption.
> > >  - The action prediction accuracy on the y-axis of Figure 5 is averaged over all 250 timesteps of each user experiment, similar to the "cumulative accuracy" in Figure 2b.
> > >  - We have adjusted the language in the paper to clarify that the backspace feedback is not “implicit”, and added this limitation to the 2nd paragraph of the discussion in Section 5.
> > >  - Thank you for the positive feedback about our experiments!
> > >  - We have updated the paper to address all other suggested edits, including changing the acronym to X2T.

---

### Official Review · AnonReviewer2 · 2020-10-29
**Interesting problem motivation.**

**Rating:** 4
**Confidence:** 4

**Review:**

This paper presents Machine Learning based approaches that map inputs to desired actions with applications to assistive interfaces. The paper leverages user feedback to improve the predictive model to perform desired actions. The paper conducts user studies that gauge the impact of the proposed interface in contrast to a non-adaptive baseline interface. The problem is well motivated and the paper is well written.

Some questions/comments:

[1] What the paper refers to as “implicit” feedback, i.e., in the context of the paper, the backspace command is actually explicit. It is clear that the assumptions enforced by the paper indicate that a backspace means a strongly negative signal, and anything that has not been backspaced to be the right action. This can, for e.g., be used for training an appropriate reward regressor. This feedback however is incomplete, because, we tend to observe the feedback only for the actions presented by the interface.

[2] This paper attempts to solve a problem that is highly non-trivial when the cardinality of actions grows large, for e.g., the English language vocabulary. It is unclear how one can utilize a standard yes/no feedback using a backspace key to effectively learn a highly accurate policy that presents us with the next action (for example, a word) given the context. The paper however doesn’t present a discussion surrounding these challenges, which for me is a major shortcoming of this paper.

[3] Naturally, some of the assumptions considered with regards to the use of backspaces is pretty strong. The feedback model assumed by the paper is essentially noiseless in that the user always does the right thing with regards to presenting feedback on good/bad actions. The paper doesn’t validate how errors in the use of the backspace key manifest themselves in the performance of the policy learning step.

---

> ### Author Response · Authors · 2020-11-14
> **Response to R2**
>
> Thank you for the thoughtful and constructive feedback. Based on the review, the main reservations about our work appear to be (1) that XT2 may be sensitive to noisy reward signals, and (2) that XT2 may not be able to effectively explore a large action space. We address (1) by adding a new experiment with the eye gaze data (Figure 5 in the appendix), and address (2) by clarifying our handwriting experiments (Sections 4.3 and 4.4). In light of these revisions and clarifications, we believe that most of these issues should be addressed. Please let us know if any further concerns remain, or if you would like to see any further modifications.
>
> Sensitivity to noisy rewards
> --------------------------------------
> To address concerns from R2 and the other reviewers about XT2’s sensitivity to noisy rewards, we have added a new experiment (Figure 5 in the appendix). We ran an experiment with the eye gaze data in which we flipped each reward r to 1-r with probability p, where p is the “reward mislabeling rate”. We found that XT2 outperformed the default interface for a wide range of mislabeling rates p, and only performed worse than the default interface when the rewards were completely random (i.e., the mislabeling rate is p=0.5). In practice, users' backspaces tend to follow the assumptions in Section 2.1 -- i.e., users tend to backspace incorrect actions, and not backspace correct actions -- leading to the low empirical mislabeling rate of p=0.014 (as mentioned at the end of Section 4.1).
>
> Exploring large action spaces
> -----------------------------------------
> R2 is correct that exploring a large action space with bandit feedback alone is challenging. To study this problem, we conducted the handwriting experiments in Sections 4.3 and 4.4 with a 26-dimensional action space of characters. To guide exploration and improve initial performance, we use a language model, as described in the 2nd paragraph of Section 4.3. The ablation studies in Section 4.4 show that incorporating the language model into our prior policy, as well as training our reward model on online user feedback, both substantially improve XT2’s accuracy. This result (see Figure 4 for details) suggests that XT2 can be applied to practical problems with large action spaces. We have added implementation details for the language model to Appendix A.1.2 in the updated paper.
>
> Implicit vs. explicit feedback
> ---------------------------------------
> We will adjust the language in the paper to clarify that backspaces are explicit feedback, rather than implicit feedback. R2 is correct that XT2 trains a reward regressor, and that bandit feedback is only given for actions taken by the interface.

---

> > ### Comment · AnonReviewer2 · 2020-11-23
> > **Comments on the author response**
> >
> > Thank you once again for the detailed responses.
> >
> > My point about large action spaces is not just towards scaling up the interface to the level of characters of English language. The real challenge is in scaling up the interface to the word level; this really does require using both a language model and (potentially) a non-trivial exploration strategy. And, in reality, we do have well-performing systems in practice, for e.g.,  look at the Gmail Smart Compose: Real-Time Assisted Writing paper by Chen et al.  (https://arxiv.org/pdf/1906.00080.pdf). This paper deals with a system that presents non-trivial challenges.  Considering noisy feedback at the scale of the system in the Gmail smart compose paper would be highly non-trivial; even with realizable rewards, when working with lesser data/badly performing model, one might require to use a sophisticated exploration strategy, for e.g. see a recent paper of Xu et al. (https://arxiv.org/pdf/2003.12699.pdf) to do gap dependent exploration.
> >
> > While I do understand the general premise of this problem setting/motivation is interesting and useful, the scale of problems considered by this paper in my view makes the solution very simple - in fact, I am not sure which of these techniques are novel excepting, perhaps their application to this problem. I am happy to understand the perspective of the authors if they believe I have missed out on anything with regards to evaluating the novelty of the algorithms that their paper presents.

---

> > > ### Author Response · Authors · 2020-11-23
> > > **Clarifications**
> > >
> > > Thank you for your response. There seem to be two concerns in your post: (1) that practical interfaces have to operate at the word level, and (2) that our proposed algorithm is not novel.
> > >
> > > Choice of action space
> > > -------------------------------
> > > While the Gmail Smart Compose method does indeed operate at the word level, it is solving a different problem than assistive typing: Smart Compose seeks to predict the user’s next words given their previous words, while in the assistive typing domain the aim is to translate the user’s input (e.g., eye gaze from a webcam, or handwriting from a stylus) into their desired word or character. Smart Compose requires access to ground-truth action labels (similar to prior work discussed in the 1st paragraph of Section 3 of our paper) in the form of a large dataset of user-composed emails, whereas our method learns online from binary feedback signals that are easier for motor-impaired users to provide than ground-truth actions. Furthermore, user input in assistive typing tends to be noisy (e.g., due to variations in webcam images), which makes predicting word-level actions more difficult than in the Smart Compose setting. Hence, while defining the action space to be a large set of words is appropriate for Smart Compose, it is not necessarily the best choice of action space for assistive typing systems. For example, the state of the art in assistive typing via brain-computer interfaces predicts individual characters, not words [1].
> > >
> > > Novelty
> > > ---------
> > > While our proposed method is based on prior work on contextual bandits, our main contribution is demonstrating its successful application to assistive typing. Furthermore, we contribute new experimental insights in the assistive typing domain on the importance of offline pretraining of the reward model, using a prior policy to guide exploration, and using online learning to overcome calibration problems (e.g., with the gaze tracker) that would otherwise cause performance to degrade over time. In addition to validating these components of our method, our experiments illustrate surprising and useful results, such as user co-adaptation to the system.
> > >
> > > [1] https://www.biorxiv.org/content/10.1101/2020.07.01.183384v1

---

> > > > ### Comment · AnonReviewer2 · 2020-11-23
> > > > **Response to author comments**
> > > >
> > > > Thank you for your comments.
> > > >
> > > > With regards to the smart compose paper, I pointed to an example system that can scale up to the next word prediction. How does it connect with this paper? Well, consider similar settings as ones explored by this paper, where, we replace character prediction with next word prediction. While the authors suggest that this doesn't fit in within current state of the art of assistive typing systems -- my point is this can be furthered by relying on potentially more advanced approaches that have appeared in the literature on contextual bandits and related areas. Otherwise, as it stands, clearly, as the authors point out, the techniques used in the paper aren't novel, their application to the specific problem potentially is - and I will leave it to the AC to judge the significance of this contribution. My broader point also is that these techniques do not appear to be promising for problems going beyond a stylized instantiation of the assistive typing problem where we predict the next character (i.e. when very deal with very low cardinality action space) conditioned on context.
> > > >
> > > > With regards to the binary feedback: I'd like to point out that the feedback mechanism is clearly supervised - backspaces implies negatives, and non-backspaces implies correct completions. This isn't implicit feedback by any means. Yes, we do observe feedback only for the actions picked by the interface, but the feedback essentially reveals the right/wrong answer.
> > > >
> > > > I will retain my current score for the submission and thank the authors for their perspective.

---

> > > > > ### Author Response · Authors · 2020-11-23
> > > > > **Follow up**
> > > > >
> > > > > Thank you for your response.
> > > > >
> > > > > We appreciate that Smart Compose is an example of a system that predicts the next word given previous words, but **one of the reasons it scales to action spaces containing thousands of words is that it is solving a different problem and makes different assumptions than the assistive typing domain we study**: Smart Compose (1) trains a model that operates on discrete token sequence inputs, rather than noisy inputs like eye gaze or handwriting; and (2) trains a model with supervised learning on ground-truth action labels, rather than bandit feedback. Precise user inputs and action labels enable Smart Compose to scale, but they are not available in typical assistive typing domains in which users can only provide noisy inputs and action feedback. In these settings, choosing an action space with thousands of words is infeasible, which is why state-of-the-art systems, e.g., for typing via brain-computer interfaces [1], often use character-level actions. In such prior work [1], it is extremely difficult for the user to encode their desired word (out of thousands of possible words) in their input, but much easier for them to encode their desired character (out of 31 possible characters). **The size of the action space is limited not just by the ability of our learning algorithm, but also by the ability of the user to provide precise inputs.**
> > > > >
> > > > > We would like to highlight that papers that focus on applications are within the scope of ICLR, and that while our proposed algorithm is not necessarily novel, our experiments show novel results (see “Clarifications” above).
> > > > >
> > > > > We agree that backspace feedback is explicit, not implicit, and have updated the paper to not use “implicit” when describing user feedback.
> > > > >
> > > > > We agree that positive feedback indicates that the interface took the correct action, but the key challenge in the bandit setting involves learning from trial and error, which includes learning from negative feedback. During learning, our method does not have access to ground-truth action labels, which means it must explore different actions and learn from both negative and positive feedback. Hence, action feedback is a weaker learning signal than action labels.
> > > > >
> > > > > [1] https://www.biorxiv.org/content/10.1101/2020.07.01.183384v1

---

> > > > > > ### Comment · AnonReviewer2 · 2020-11-25
> > > > > > **Response to author comments**
> > > > > >
> > > > > > Thank you for your comments. Just to ensure we are on the same page - I do believe applications are (an integral) part of ICLR.
> > > > > >
> > > > > > That said, despite the authors response, I do retain my reservations about the scope of contributions of this paper (which of course deals with an interesting paradigm), and, as I have made clear, I do believe this paper requires more work to be accepted for publication. I will leave it to the AC to make the final call on this paper.

---

> ### Author Response · Authors · 2020-11-20
> **Request for Feedback**
>
> Thank you again for the thoughtful review. We would like to know if our rebuttal (see below, "Response to R2”) adequately addressed your concerns. We would also appreciate any additional feedback on the revised paper. Are there any other aspects of the paper that you think could be improved?

---

> ### Author Response · Authors · 2020-11-22
> **Please Respond to Author Rebuttal**
>
> **The discussion period ends in two days**, and we would like to have enough time to run any additional experiments that you might suggest. We would like to know if our rebuttal (see below, "Response to R2”) adequately addressed the concerns in your original review. We would also appreciate any additional feedback on the revised paper. Are there any other aspects of the paper that you think could be improved?

---

### Decision · Program_Chairs · 2021-01-07
**Final Decision**

**Decision:**

Accept (Poster)

**Comment:**

The paper describes a cool application of online learning from bandit feedback -- creating personalized, adaptive typing interfaces for users with sensorimotor impairments. The problem is well-motivated -- the interface can observe users' gaze (e.g. via a webcam image), predict a character as an action, and bandit feedback can be collected by observing whether users use the backspace key after the interface's action. Prior work showed that gaze-to-text can be less burdensome than typing, but this can quickly become untrue the more mistakes the interface makes. So, the goal is to personalize the interaction policy so that it makes fewer mistakes than the default interaction policy trained using a fixed dataset of expert demonstrations.

The high-point of the paper is the empirical user study with 12-60 participants -- the study convincingly demonstrates that indeed a simple bandit algorithm can improve over the default interface; moreover, users exhibit intriguing co-adaptation patterns with the adaptive interfaces. These findings may prove to be an interesting point for future studies in user co-adaptation.

The low-point of the paper is its algorithmic development. There is a vast literature on bandit/RL algorithms, and incorporating human feedback into their operation (the paper rightly cites TAMER, COACH, etc.) but it is very unclear why any one of these algorithms could not be used for the paper's application. COACH (human feedback gives an explicit view of the action's advantage -- which in the contextual bandit setting exactly matches the paper's assumptions) seems particularly appropriate. Although the algorithm proposed in the paper is simple, how applicable is it in any other context? how does it compare to COACH/etc.? when should we prefer this algorithm over others? Furthermore, given that X2T trains a reward model from observed user-behavior, a natural baseline would use an epsilon-greedy strategy (fraction of the time, pick actions greedily according to the reward model) -- this might isolate the benefit of the approximately Boltzmann exploration being conducted on top of the reward estimates in Eqn 2. Finally, since X2T trains a reward model per user it could be particularly informative to visualize what the models have learned to illustrate qualitatively how X2T is personalizing across its user base.

The paper could have a much bigger impact if the authors can figure out some creative way to enable the broader research community to work on this problem domain. A testbed or environment (like RecSim for content recommendation https://github.com/google-research/recsim) with configurable but realistic reward models could allow researchers to test several bandit algorithms, MDP vs CB formulations, other ways to interpret user feedback etc.